# Insulin Complexation with Cyclodextrins—A Molecular Modeling Approach

**DOI:** 10.3390/molecules27020465

**Published:** 2022-01-11

**Authors:** Pálma Bucur, Ibolya Fülöp, Emese Sipos

**Affiliations:** 1Department of Drugs Industry and Pharmaceutical Management, Faculty of Pharmacy, George Emil Palade University of Medicine, Pharmacy, Science and Technology of Târgu Mures, 540142 Targu Mures, Romania; bucur_palma@yahoo.com (P.B.); emese.sipos@umfst.ro (E.S.); 2Department of Toxicology and Biopharmacy, Faculty of Pharmacy, George Emil Palade University of Medicine, Pharmacy, Science and Technology of Târgu Mures, 540142 Targu Mures, Romania

**Keywords:** insulin, amyloid fibrillation, hydroxypropyl-β-cyclodextrin, sulfobutylether-β-cyclodextrin, complex, AutoDock, molecular modeling, docking

## Abstract

Around 5% of the population of the world is affected with the disease called diabetes mellitus. The main medication of the diabetes is the insulin; the active form is the insulin monomer, which is an instable molecule, because the long storage time, or the high temperature, can cause the monomer insulin to adapt an alternative fold, rich in β-sheets, which is pharmaceutically inactive. The aim of this study is to form different insulin complexes with all the cyclodextrin used for pharmaceutical excipients (native cyclodextrin, methyl, hydroxyethyl, hydroxypropyl and sulfobutylether substituted β-cyclodextrin), in silico condition, with the AutoDock molecular modeling program, to determine the best type of cyclodextrin or cyclodextrin derivate to form a complex with an insulin monomer, to predict the molar ratio, the conformation of the complex, and the intermolecular hydrogen bonds formed between the cyclodextrin and the insulin. From the results calculated by the AutoDock program it can be predicted that insulin can make a stable complex with 5–7 molecules of hydroxypropyl-β-cyclodextrin or sulfobutylether-β-cyclodextrin, and by forming a complex potentially can prevent or delay the amyloid fibrillation of the insulin and increase the stability of the molecule.

## 1. Introduction

According to the World Health Organization, over 422 million people are affected with the disease called diabetes mellitus [1]. There are mainly two types of diabetes: type 1 is juvenile diabetes, an autoimmune disorder, often referred to as insulin-dependent diabetes mellitus; and type 2 is adult-onset diabetes, which results from defective insulin secretion, and can be insulin or non-insulin dependent. Insulin is the main medication used in the treatment of diabetes mellitus [2,3,4].

The insulin is a peptide hormone, secreted by the pancreas, first discovered in 1922 by Banting and Macleod [5]. This is a small protein containing 51 amino acids (17 of the proteinogenic amino acids), composed of 2 polypeptide chains: A and B. Chain A contains 21 amino acids and these are forming two small α-helixes between A1–A7 and A12–A18 residues. Chain B contains 30 amino acids, and these are forming a larger α-helix between B8–B23 fragments. In Figure 1, it is depicted the 3 α-helixes of the insulin. From the 51 residues, 6 of them are cysteine, which form 3 disulfide bonds inside the insulin. Two of these bonds make a linkage between the two chains (A7–B7 and A20–B19) and one is making a bond between the two parts of the A chain (A6–A11). These disulfide bonds are helping to maintain the protein’s secondary structure [6,7].

The insulin active form is a monomer (containing an A and a B chain). These monomers can easily associate into a dimer form (containing chains A, B, C and D), and later to a tetramer and hexamer form. These hexamers can make a stable complex by binding two Zn^2+^ ions (or other bivalent metal ions). Figure 2 shows a stable hexamer complex with two Zn^2+^ ions.

In the pharmaceutical forms of insulin, it is often stored as a hexamer complex with two bivalent ions (ex. Zn^2+^ or Ca^2+^) to prevent the fibrillation of the molecule, because the high temperature, or the long storage time can cause the monomer insulin to adapt an alternative fold, rich in β-sheets, which is pharmaceutically inactive [6,10].

The formation of the amyloid fibrillation can cause many problems during the manufacturing process, in the delivery or storage stage. There are many attempts to stabilize the molecule in the dimer (active) form. One of these methods is to form a complex with some excipients, such as cyclodextrins. Some of the cyclodextrins, such as hydroxypropyl-β-cyclodextrin, sulfobutylether-β-cyclodextrin or 6-*O*-α-(4-*O*-α-d-glucuronyl)-d-glucosyl-β-cyclodextrin, are proved to be capable to delay or prevent the amyloid fibril formation in vitro condition [11,12].

The cyclodextrins are cyclic oligosaccharides, with 6, 7 or 8 glucopyranose units, attached by α (1–4) glucoside bonds, and named α, β and γ cyclodextrin, respectively. They have a truncated conelike form, which differs in the ring size, depending on the number of glucose units. The internal surface of the cavity of the cyclodextrin is hydrophobic and the exterior is hydrophilic, because of the arrangement of the hydroxyl groups in the molecule. Thanks to those properties, the cyclodextrin is capable to form inclusion complexes with other small hydrophobic molecules or with a hydrophobic part of a bigger molecule. By using these excipients, we can improve the solubility, stability, bioavailability, taste, odor and other physical properties of the pharmaceutically active ingredients [13,14,15,16]. Native cyclodextrins are α, β and γ cyclodextrin. By substitution of one or more H of hydroxyl groups on the glucopyranose unit (in the position 2, 3 or 6) with methyl, ethyl, hydroxyl or other functional groups (ex. sulfobutylether), cyclodextrin derivatives can obtain. In the Handbook of Pharmaceutical Excipient [13], eight types of cyclodextrin are listed, as used in pharmaceutical formulation or technology: α-cyclodextrin, β-cyclodextrin, γ-cyclodextrin, dimethyl-β-cyclodextrin, trimethyl-β-cyclodextrin, 2-hydroxyethyl-β-cyclodextrin, 2-hydroxypropyl-β-cyclodextrin and sulfobutylether-β-cyclodextrin. These cyclodextrins are used in oral, parenteral, topical and ophthalmic pharmaceutical formulations. The β-cyclodextrin can cause severe nephrotoxicity in parenteral formulation, but the other cyclodextrins are considered safe to use in any kind of pharmaceutical formulation. The β-cyclodextrin derivatives have a greater water solubility (1 in less than 2 parts of water at 25 °C), and more favorable complexing properties than the β-cyclodextrin. Moreover, since they are not nephrotoxic, it is safe to use them in parenteral formulation [13,17].

The aim of this study is to find the best type of cyclodextrin (in silico conditions) for stabilizing the insulin molecule in the active form. For this purpose, a searching algorithm, provided by Scripps Research Institute, in the AutoDock molecular modeling program was used. AutoDock is an automated molecular modeling software, which has been available for the public from 1990 onward, with a GNU General Public License [18]. More than 7000 publications are registered on the PubMed Central, using the AutoDock for a docking method and over 30,000 citations are reported by Google Scholar [19]. It is a fast, atom-based, computational docking tool, which can determine the best position of the ligand in the active site of the peptide molecule by exploring the translation, orientation and conformation of the ligand, until the smallest bonding energy of the complex is found. This program can use three kinds of algorithms: Monte Carlo simulated annealing, genetic algorithm and Lamarckian genetic algorithm. To provide a greater efficiency, all of these algorithms are using a rapid, grid-based method for the energy evaluation [20,21,22,23]. These energies are calculated by the master equation, which contains a model for the dispersion repulsion, hydrogen bonding, electrostatics, torsion and desolvation energy. The dispersion repulsion energy is calculated by the Lennard-Jones function, it helps to calculate the energy of the Van der Waals bonds, and to preserve the radius of the atoms in the simulated space. The energy of the hydrogen bond is also calculated using the Lennard-Jones function, substituting different constants, hence managing to calculate the length and the angle of the hydrogen bonds [20,23,24,25]. The electrostatic energies are determined by a Coulomb potential screening, using a precalculated grid map. The torsion energy is proportionate with the *sp*^3^ bonds, and determines a restriction for the degree of freedom based on the entropy of the molecules. The evaluation of the solvation energy is based on the accessible surface of the molecule, taking into account the type of atoms and the dielectric constant of these atoms [20,21,23]. With this precise equation, the bonding energy and the conformation of the peptide complex can be determined, and by comparing results of two or more complexes with each other, the best complex with the greater stability can be predicted.

Another aim of this study is to form different insulin complexes with all the cyclodextrin used for pharmaceutical excipients (according to the Handbook of Pharmaceutical Excipient [13]) in silico condition, to determine the best type of cyclodextrin or cyclodextrin derivate to form a complex with an insulin monomer, to predict the molar ratio, the conformation of the complex, and the intermolecular hydrogen bonds formed between the cyclodextrin and the insulin. For predicting the molar ratio and the conformation of the complex, we designed an accurate, cost efficient procedure to dock the insulin with more than one ligand (theoretically between 5–10, according to the literature [11]), but we also kept the computational demands at a reasonable level. This method is detailed in the Method section.

## 2. Results 

As the result of blind docking (executed in the CB-Dock) for every cyclodextrin from 5 to 9 grid parameter coordinates were obtained. Although there were 18 different sets of blind docking (with 10 positions) computed, only 9 specific coordinates on the insulin molecule were obtained. The center of these grid parameters is shown on the Figure 3.

The result of the long docking process is detailed in Table 1. According to the binding energy, the number of cyclodextrins, which can bind to an insulin molecule were determined. Because the AutoDock program has a standard error around 2.5 kcal/mol [19], and a bonding energy greater than −1 kcal/mol is too weak to form a complex, only those cyclodextrin were taken in consideration to participate in the complex with insulin, which has a smaller bonding energy, than −3.5 kcal/mol.

According to the results, the insulin forms a complex with 8 molecules of α-cyclodextrin, or 7 molecules of β-cyclodextrin, or 5 molecules of γ-cyclodextrin. The insulin does not form a complex with the methyl or the hydroxyethyl substituted β-cyclodextrin, except with the oligomethyl-β-cyclodextrin. The insulin forms a complex with the 5 to 7 molecules of hydroxypropyl substituted β-cyclodextrin or sulfobutylether substituted β-cyclodextrin, depending on the size and the degree of substitution of the respective cyclodextrin.

### 2.1. Insulin Complexes with Native Cyclodextrins

#### 2.1.1. Insulin Complexes with α-Cyclodextrin

In the insulin α-cyclodextrin complex 8 molecules of α-cyclodextrin are arranged around the insulin molecule, with an average bonding energy of −8.55 kcal/mol. There are four residues on the insulin, which are entering in four different α-cyclodextrin cavities: A:GLN5, B:PHE1, B:PHE25, B:LYS29 (Figure 4). The AutoDock program detected 22 hydrogen bonds between the insulin and the cyclodextrins, and from these, 8 bonds in the A chain (with residue: A:GLY1, A:GLY5, A:SER12, A:GLN15, A:TYR19, A:ASN21) and 14 bonds in the B chain (with residue: B:PHE1, B:ASN3, B:GLY8, B:SER9, B:ARG22, B:PHE24, B:TYR26, B:LYS29, B:THR30). The best binding energy was in position 7, −10.11 kcal/mol, around the residue B:LYS29.

#### 2.1.2. Insulin Complexes with β-Cyclodextrin

In the insulin β-cyclodextrin complex comprises 7 molecules of β-cyclodextrin bonded to the insulin molecule, with an average binding energy of −9.43 kcal/mol. In six different β-cyclodextrin cavities entered the following insulin residues: A:LEU13, A:TYR14, B:CYS7, B:GLU21, B:LYS29 and in the same cavity B:SER9, B:HIS10, B:GLU13 (Figure 5). The AutoDock program detected 25 hydrogen bonds between the insulin and the cyclodextrins, from this 6 bonds in the A chain (with residue: A:THR8, A:TYR14, A:GLN15, A:ASN18, A:ASN21) and 19 bonds in the B chain (with residue: B:VAL2, B:ASN3, B:GLN4, B:HIS5, B:CYS7, B:SER9, B:HIS10, B:LEU11, B:ARG22, B:GLY23, B:PHE24, B:TYR26, B:LYS29, B:THR30). The best bonding energy was in the position 6, −10.3 kcal/mol, around the residue B:CYS7.

#### 2.1.3. Insulin Complexes with γ-Cyclodextrin

In the insulin γ-cyclodextrin complex, 5 molecules of γ-cyclodextrin are arranged around the insulin molecule, with an average bonding energy of −7.89 kcal/mol. In three different γ-cyclodextrin cavities entered the next insulin residues; in one cavity A:TYR14 and A:GLN15; in another, cyclodextrin B:SER9, B:HIS10, and B:GLU13; and in the last cavity, B:TYR16 (Figure 6). The AutoDock program detected 16 hydrogen bonds between the insulin and the cyclodextrins, and from these, 8 bonds in the A chain (with residue: A:GLN5, A:CYS7, A:SER9, A:TYR14, A:GLN15, A:ASN18, A:TYR19, A:ASN21) and 8 bonds in the B chain (with residue: B:GLN4, B:GLY8, B:SER9, B:HIS10, B:LUE15, B:TYR16, B:TYR26, B:THR27). The best bonding energy was in the position 2, −9.73 kcal/mol, around the residue A:TYR14.

### 2.2. Insulin Complexes with Methyl Substituted Cyclodextrins

#### 2.2.1. Insulin Complexes with Methyl-α-Cyclodextrin (DS 12 and 18)

In the case of methyl-α-cyclodextrin (with degrees of substitution of 12 and 18), all binding energy was greater than −3.5 kcal/mol, in all the simulated position. The average simulated bonding energy for methyl-α-cyclodextrin, with a degree of substitution of 12, was −0.97, and the best results were in position 3, with −2.38 kcal/mol, around the residue B:LYS29. For the methyl-α-cyclodextrin, with a degree of substitution of 18, the average energy was −0.89 kcal/mol, with the best result of −3.06 kcal/mol, in position 3, around the B:PHE1 residue. Consequently, none of the methyl-α-cyclodextrin can form a complex with the insulin molecule.

#### 2.2.2. Insulin Complexes with Methyl-β-Cyclodextrin (DS 6)

With the methyl-β-cyclodextrin, and a degree of substitution of 6, there formed a complex with 5 molecules of methyl-β-cyclodextrin, with an average bonding energy of −6.06 kcal/mol. In five different methyl-β-cyclodextrin cavities, the next insulin residues were entered; in one cavity A:GLN5, A:GLN15 and A:ASN18 were entered; in another cavity, B:GLY8 and B:SER9 were entered; and in different cavities, one by one the residues enumerated as follows were entered: B:PHE1, B:TYR16 and B:LYS 29 (Figure 7). The AutoDock program detected 16 hydrogen bonds between the insulin and the cyclodextrins, and from these, 6 bonds in the A chain (with residue: A:GLY1, A:ILE2, A:GLN5, A:ASN18 and A:TYR19) and 10 bonds in the B chain (with residue: B:PHE1, B:GLN4, B:GLY8, B:SER9, B:HIS10, B:PHE24, B:TYR26, B:LYS29, B:THR30). The best binding energy was in position 4, −8.59 kcal/mol, around the residue B:LYS29.

#### 2.2.3. Insulin Complexes with Methyl-β-Cyclodextrin (DS 12)

The methyl-β-cyclodextrin, with a degree of substitution of 12, had a binding energy greater than −3.5 kcal/mol, in all the simulated positions. The average energy were 1.13 kcal/mol, with the best binding energy of 0.05 kcal/mol, around the residue A:ASN18. In conclusion, the methyl-β-cyclodextrin, with a degree of substitution of 12, does not bond in a complex with the insulin molecule.

### 2.3. Insulin Complexes with Hydroxypropyl Substituted Cyclodextrin Complexes

#### 2.3.1. Insulin Complexes with Hydroxypropyl-β-Cyclodextrin (DS 1)

The hydroxypropyl-β-cyclodextrin, with a degree of substitution of 1, can form a stabile complex with insulin, containing 7 molecules of hydroxypropyl-β-cyclodextrin, with an average binding energy of −10.89 kcal/mol. The best binding energy was −12.83 kcal/mol, in position 4, located around the residue A:LEU13. In five different cyclodextrin cavities, some residues from the insulin were entered; B:PHE1 and B:GLN4 entered in one cavity; B:LYS29 and B:THR30 entered into another cavity and the next three residues were located in three different cyclodextrin: A:LEU13, A:TYR14 and B:TYR16 (Figure 8). The AutoDock program detected 25 hydrogen bonds between the insulin and the cyclodextrins, and from these, 6 were in the A chain (with residues: A:GLY1, A:GLN5, A:LEU13, A:ASN18, A:ASN21) and 19 were in the B chain (with residues: B:PHE1, B:VAL2, B:ASN3, B:GLN4, B:HIS5, B:CYS7, B:SER9, B:HIS10, B:TYR16, B:GLU21, B:PHE24, B:TYR26 and B:LYS29).

#### 2.3.2. Insulin Complexes with Hydroxypropyl-β-Cyclodextrin (DS 2)

The hydroxypropyl-β-cyclodextrin, with a degree of substitution of 2, were forming a stable complex with insulin, containing 7 molecules of hydroxypropyl-β-cyclodextrin located around the next residues: A:LEU13, B:HIS10, B:TYR16, and B:LYS29, and in one cavity A:GLN5 with A:ASN18 were entered (Figure 9). The average binding energy was −5.46 kcal/mol, and the best bonding energy was −8.49 kcal/mol, in position 6, around the residue B:LYS29. The AutoDock program detected 20 hydrogen bonds between the insulin and the cyclodextrins, and from these, 3 were on chain A (with residues: A:GLN5, A:CYS7, and A:TYR14) and on chain B were 17 (with residues: B:PHE1, B:CYS7, B:GLY8, B:SER9, B:HIS10, B:GLY20, B:ARG22, B:GLY23, B:TYR26, B:THR27, B:LYS29 and B:THR30).

#### 2.3.3. Insulin Complexes with Hydroxypropyl-β-Cyclodextrin (DS 3)

The hydroxypropyl-β-cyclodextrin, with a degree of substitution of 3, can form a stabile complex with insulin, containing 6 molecule of hydroxypropyl-β-cyclodextrin, with an average bonding energy of −10.89 kcal/mol, the best bonding energy being −9.06 kcal/mol, in position 4, around the residue B:LYS29. In 5 of the cyclodextrin cavities, the next residues entered from the insulin molecule: A:GLN5, B:PHE1, B:CYS7, B:TYR16 and B:LYS29 (Figure 10). The AutoDock detected 24 hydrogen bonds between the insulin and the cyclodextrins, and from this, 4 were on the A chain (with residues: A:GLN5, A:THR8, A:SER9, A:TYR14), and 19 on the B chain (with residues: B:PHE1, B:VAL2, B:ASN3, B:GLN4, B:HIS5, B:CYS7, B:SER9, B:HIS10, B:TYR16, B:PHE24, B:LYS29 and B:THR30).

#### 2.3.4. Insulin Complexes with Hydroxypropyl-β-Cyclodextrin (DS 4)

The hydroxypropyl-β-cyclodextrin, with a degree of substitution of 4, is capable of forming a complex with the insulin molecule, with 6 molecules of hydroxypropyl-β-cyclodextrin. These cyclodextrins incorporate the next residues of the insulin in their cavities: A:LEU13; A:TYR14 and B:GLN15 in one cavity; B:SER9 and B:HIS10 in the second cavity; B:GLU21 and B:ARG22 in the third cavity; B:PRO28 and B:LYS29 in another cavity; and B:THR30 in the last cavity (Figure 11). The average binding energy of the complex is −5.09 kcal/mol, with the best energy of −6.82 kcal/mol, in position 1, at the residues B:PRO28 and B:LYS29. The AutoDock program detected 17 hydrogen bonds between the insulin and the cyclodextrins, and from these, 7 were on the A chain (with residues: A:ILE2, A:GLN5, A:SER9, A:TYR14, A:TYR19 and A:CYS20), and 10 were on the B chain (with residues: B:ASN3, B:GLN4, B:SER9, B:HIS10, B:TYR16, B:ARG22 and B:LYS29).

### 2.4. Insulin Complexes with Sulfobutylether Substituted Cyclodextrins

#### 2.4.1. Insulin Complexes with Sulfobutylether-β-Cyclodextrin (DS 1)

The sulfobutylether-β-cyclodextrin, with a degree of substitution of 1, can form a stable complex with an insulin molecule, containing 7 molecules of sulfobutylether-β-cyclodextrin, with the average bonding energy of −8.49 kcal/mol, the best energy being −12.15 kcal/mol, in position 6, around the residue A:GLN5 and A:GLN15. In the case of sulfobutylether-β-cyclodextrin, only 3 residues enter in the cavities of the cyclodextrin (A:GLN5, A:LEU13 and B:THR30), and in the rest of the position the cyclodextrin do not take in the residues in their cavities, instead arranging really closely to the insulin molecules, with the sulfobutylether group facing the insulin (Figure 12). The AutoDock program detected 30 hydrogen bonds between the insulin and the cyclodextrin molecules, and from these, 14 were on chain A (with residues: A:GLY1, A:GLN5, A:THR8, A:SER9, A:SER12, A:LEU13, A:TYR14, A:GLN15, A:ASN18, A:TYR19 and A:ASN21) and 16 were on chain B (with residues: B:PHE1, B:VAL2, B:ASN3, B:GLN4, B:HIS5, B:GLY8, B:HIS10, B:TYR16, B:TYR26, B:THR27, B:LYS29 and B:THR30).

#### 2.4.2. Insulin Complexes with Sulfobutylether-β-Cyclodextrin (DS 2)

In the case of sulfobutylether-β-cyclodextrin, with a degree of substitution of 2, with an average binding energy of −12.96 kcal/mol, the cyclodextrin making a stable complex with the insulin, using 6 molecules of sulfobutylether-β-cyclodextrin (Figure 13), there were none of the residues entered into the cavities of the cyclodextrins, but were located very close to the next residues, facing the sulfobutylether part to the insulin: A:VAL3, A:GLN5, A:LEU13, B:PHE1, B:LEU6, B:CYS7, B:GLY8, B:HIS10, B:GLU13, B:ALA14, B:LEU17, B:GLU21, B:GLY23, B:THR27, B:PRO28, B:LYS29 and B:THR30.

The best binding energy was −18.70 kcal/mol, in position 6, around the residue B:LYS29 and B:THR30. The AutoDock program detected 21 hydrogen bonds between the insulin and the cyclodextrin molecules, and from these, 7 were on chain A (with residues: A:GLY1, A:GLN5, A:ASN18, A:TYR19 and A:ASN21) and 14 were on the chain B (with residues: B:PHE1, B:VAL2, B:ASN3, B:GLN4, B:GLY8, B:SER9, B:HIS10, B:LYS29 and B:THR30).

#### 2.4.3. Insulin Complexes with Sulfobutylether-β-Cyclodextrin (DS 3)

The sulfobutylether-β-cyclodextrin, with a degree of substitution of 3, is capable to form a stable complex with the insulin, containing 5 molecules of sulfobutylether-β-cyclodextrin, with an average energy of −10.68 kcal/mol. Part of the insulin residues does not enter into the cavity of the cyclodextrin, like in the previous complexes, and instead most of the sulfobutylether groups point toward the insulin molecule, to the next residues: A:CYS7, A:LEU13, B:LEU6, B:CYS7, B:GLY8, B:TYR16, B:VAL18, B:GLU21, B:GLY23, B:PHE24, B:TYR26, and B:PRO28 (Figure 14). The best bonding energy was −16.29 kcal/mol, in position 6, around the residues B:TYR16 and B:GLU21. The AutoDock detected 27 hydrogen bonds between the insulin and cyclodextrin molecules, and from these, 8 were on chain A (with residues: A:GLY1, A:GLN5, A:THR8, A:LEU13, A:TYR14, A:LEU16, A:ASN21), and 19 were on chain B (with residues: B:PHE1, B:VAL2, B:ASN3, B:GLN4, B:HIS5, B:SER9, B:HIS10, B:TYR16, B:ARG22, B:TYR26, B:LYS29 and B:THR30).

### 2.5. Insulin Complexes with Hydroxyethyl Substituted Cyclodextrins

The average simulated binding energy in the case of hydroxyethyl-β-cyclodextrin, with a degree of substitution of 1, 2 or 3, was over −3.5 kcal/mol for all positions. The average energy of the hydroxyethyl-β-cyclodextrin, with a degree of substitution of 1, was −0.72 kcal/mol, with the best energy of −3.05 kcal/mol, in position 2 and around residue A:ASN18. The average binding energy of the hydroxyethyl-β-cyclodextrin, with a degree of substitution of 2, was 0.61 kcal/mol, with the best energy of −0.49 kcal/mol, in position 2, around the residue A:ASN18. The average energy of the hydroxyethyl-β-cyclodextrin, with a degree of substitution of 3, was −0.28 kcal/mol, with the best energy of −3.25 kcal/mol, in position 6, around the residues A:GLN5 and A:TYR14. In conclusion, the hydroxyethyl-β-cyclodextrin, with a degree of substitution of 1, 2 or 3, does not bond in a complex with the insulin molecule.

## 3. Discussion

Summarizing the obtained results, the insulin can form a stable complex with sulfobutylether-β-cyclodextrins, hydroxypropyl-β-cyclodextrins and native cyclodextrins. Shinde, M.N. et al. and Kitagawa, K. et al. get to the same conclusion, when they proved, that the hydroxypropyl-β-cyclodextrins, sulfobutylether-β-cyclodextrins and 6-O-a-(4-O-a-d-glucuronyl)-d-glucosyl-b-CyD can effectively prevent or delay the fibrillation of the insulin molecule [11,12].

There are a few examples in the scientific literature for the cyclodextrins to be a ligand in a docking process, Li W. et al. conducted similar research where they successfully docked a hydroxypropyl-β-cyclodextrin complex in a grape seed extract [16]. In our results, the insulin cyclodextrin complexes have an average binding energy between −5.09 kcal/mol and −12.96 kcal/mol. These findings were similar with Zor E. et al.’s research result, where they docked together some cyclodextrin with reduced graphene oxide and got a bonding energy of −9.5 kcal/mol [26]. In Mahendra V.P. et al. findings, the result of the docking was −6.1 kcal/mol between the insulin and the rutin molecule, which were used to prevent the fibrillation process [27].

In our research, most of the time, there were 5 residues on the insulin molecules, which were entered in the cavity of the cyclodextrins, and these residues were: A:LEU13, B:PHE1, B:HIS10, B:TYR16 and B:LYS29 (Figure 15).

The AutoDock program detected, on average, 22 hydrogen bonds between every insulin and cyclodextrin, in the complexes. In most of the case on the A chain, the following residues were participating in the hydrogen bond: A:GLY1, A:GLN5, A:TYR14, A:ASN18, A:TYR19 and A:ASN21 (Figure 16). These findings are in concordance with Anirban D. et al. and V.P. Mahendra et al. results, who also find some hydrogen bonds on the residues A:GLN4, A:TYR19, respectively, on A:TYR14, when they tried to stop the amyloid fibrillation of the insulin with a small molecule [27,28].

In the case of the B chain, the most commonly occurring hydrogen bonds were with the residues: B:PHE1, B:VAL2, B:ASN3, B:GLN4, B:GLY8, B:SER9, B:HIS10, B:TYR16, B:PHE24, B:TYR26, B:LYS29, and B:THR30 (Figure 17 and Figure 18).

Similar findings have Sadrjavadi K. et al., Alijanvand S.H. et al. and Kaur P. et al., respectively, they found hydrogen bond on B:VAL2, B:GLN4, B:HIS10, B:HIS10, B:TYR16, B:PHE24, B:TYR26, respectively B:TYR16, B:PHE24, B:TYR26, and B:VAL2, B:GLY8, B:HIS10, when they were locking for a stabilizing agent to prevent the amyloid fibrillation of the insulin molecule [29,30,31].

## 4. Materials and Methods 

### 4.1. Preparing the Molecules

The virtual 3D structures of the molecules were obtained from three different sources. The structure of the insulin monomer (PDB ID: 5ENA), the α-cyclodextrin (PDB ID: 1CXF), the β-cyclodextrin (PDB ID: 6JEQ) and the γ-cyclodextrin (PDB ID: 2ZYK) were downloaded from the Protein Data Bank in PDB file format [7,32,33,34,35]. The cyclodextrin derivatives were obtained from the Cambridge Structural Database, in MOL2 file format: dimethyl-α-cyclodextrin (Ref. cod: ROQVOI), trimethyl-α-cyclodextrin (Ref. cod: BEYLOG), dimethyl-β-cyclodextrin (Ref. cod: WAGHAN), oligomethyl-β-cyclodextrin (Ref. cod: IQOZIX) [36,37,38,39]. The hydroxyethyl-β-cyclodextrin, hydroxypropyl-β-cyclodextrin and sulfobutylether-β-cyclodextrin were drawn in different degrees of substitution and put under geometric optimization in HyperChem 8.0 software [40,41], to find the best conformation of the molecules, then were saved in an MOL2 file format. All the molecules were then prepared for docking by removing the extra molecules and water molecules from the files in the AutoDock program; finally, all the molecules were saved in a PDB file format. Then the molecules were transformed into PBDQT files, by being set up as ligands in the AutoDock program.

### 4.2. Cavity-Detection Guided Blind Docking

A blind docking process was carried out in order to determine the grid parameters for the molecular docking. For this purpose, the CB-Dock web server was used, which automatically identifies the binding sites of the protein molecule for a specified ligand, using the AutoDock Vina software [42,43]. When an insulin–cyclodextrin complex is formed, the molar ratio may vary from 1:1 to 1:10, and therefore, it was decided to ask for 10 cavities in the CB-Dock docking [44,45,46]. From these results, only those parameters were used where the coordinate of the cyclodextrin did not overlap with the previous cyclodextrin coordinate, and they were taken in decreasing order of the Vina score.

### 4.3. Short Docking

For flexible docking, some residues are needed to be selected to move around the ligand. In order to determine these residues, a short docking process was carried out, and all the residues which had at least one intermolecular hydrogen bond with the ligand (taking into account all 10 complexes) were used in the flexible docking as flexible residues. If there were more than 8 residues forming a hydrogen bond, then the first 8 residues were used (from the complex with the lowest binding energy to the highest binding energy). If there were less the 5 residues forming a hydrogen bond, then up to 5 residues were handpicked from all the residues which were inside the grid box, and appeared to be closest to the ligand.

The short docking process was executed in the AutoDock program, using a Lamarckian genetic algorithm, with the following searching parameters: the number of GA runs: 10; the population size: 150; the maximum number of evals: short (250,000); the maximum number of generations: 27,000; the maximum number of top individuals that automatically survive: 1; the rate of gene mutation: 0.02; the rate of crossover: 0.8; GA crossover mode: twopt; the mean of the Cauchy distribution for gene mutation: 0.0; the variance of the Cauchy distribution for gene mutation: 1.0; and the number of generations for picking the worst individuals: 10.

The parameters for the grid box were taken from the blind docking, from the CB-Dock server, but the grid box size was set to 60 × 60 × 60 or 70 × 70 × 70 (depending on the size of the cyclodextrins), because the CB-Dock underestimated the size of the ligands.

### 4.4. Flexible Docking

The flexible docking was performed in the AutoDock program with the Lamarckian genetic algorithm using the following searching parameters: the number of GA runs: 10; population size: 300; the maximum number of evals: medium (2,500,000); the maximum number of generations: 50,000; the maximum number of top individuals that automatically survive: 1; the rate of gene mutation: 0.02; the rate of crossover: 0.8; GA crossover mode: twopt; the mean of the Cauchy distribution for gene mutation: 0.0; the variance of the Cauchy distribution for gene mutation: 1.0; the number of generations for picking the worst individuals: 10.

For every ligand 5–10 dockings were performed, depending on the result obtained from the CB-Dock, and respecting the following steps:Step 1: Performing a short docking, using the insulin molecule (PDBQT file), the cyclodextrin molecule (PDBQT file), and the first cavity (according to the CB-Dock Vina score);Step 2: Performing a flexible docking, using the insulin molecule (PDBQT file), the cyclodextrin molecule (PDBQT file), and the first cavity (according to the CB-Dock Vina score), and the flexible residues determined in the short docking;Step 3: Saving the best insulin-cyclodextrin complex (PDBQT file), with the smallest binding energy.

Then the first three steps were repeated with the same cyclodextrin molecule, but instead of the insulin molecule, the best complex was used from the latest step, and for the grid box the next cavity was used (according to the CB-Dock Vina score). In that way, at every third step, the complex used as the macromolecule became bigger with a cyclodextrin molecule. These three steps were repeated, until all the cavity was utilized from the results received from the CB-Dock results, so at the end of the series of the simulation the molar ratio and the conformation of the insulin–cyclodextrin complex can be predicted.

## 5. Conclusions

In conclusion, the insulin molecule can form a complex with the native cyclodextrins, hydroxypropyl-β-cyclodextrin, sulfobutylether-β-cyclodextrin, usually in molar ratios of 1:5–7 insulin:cyclodextrin. The best complex with the insulin molecule was formed with sulfobutylether-β-cyclodextrin, with a degree of substitution of 2, and the best binding energy was −18.70 kcal/mol around the residue B:LYS29 and B:THR30. These two residues were entered in most of the cases in the cavity of the cyclodextrins, with another four insulin residues (A:LEU13, B:PHE1, B:HIS10, and B:TYR16). From the number of the hydrogen bonds appearing in the complex, and the binding energy calculated by the AutoDock program, it can be predicted that insulin can make a stable complex with 5–7 molecules of hydroxypropyl-β-cyclodextrin or sulfobutylether-β-cyclodextrin, and by forming a complex, can potentially prevent or delay the amyloid fibrillation of the insulin and increase the stability of the molecule.

## Figures and Tables

**Figure 1 molecules-27-00465-f001:**
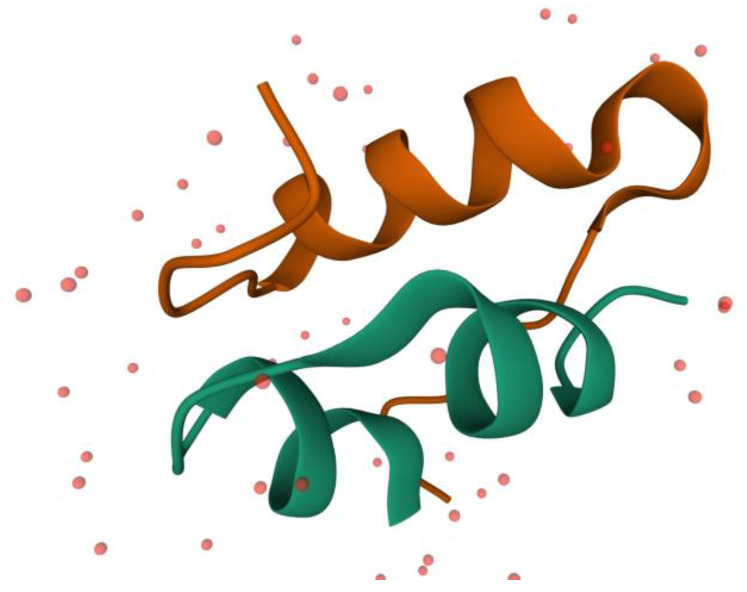
A simplified presentation of a human insulin monomer, based on the X-ray crystal structure (PDB ID: 5ENA), where only the secondary structure of the protein is shown, the A chain is marked with green and the B chain with orange, and the small red dots are the water molecules [7,8].

**Figure 2 molecules-27-00465-f002:**
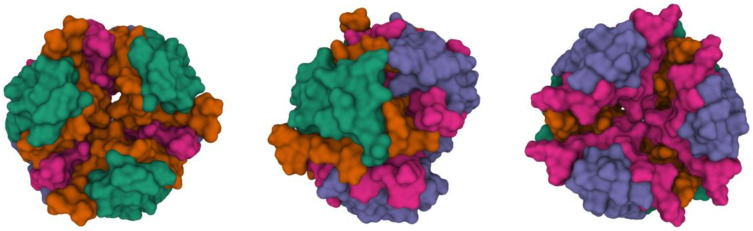
The X-ray crystal structure of a human insulin hexamer (PDB ID: 1MSO) shown from the front, side and back. In this picture the molecular surface is represented and every chain is colored with a different color (A—green, B—orange, C—purple, D—magenta) [8,9].

**Figure 3 molecules-27-00465-f003:**
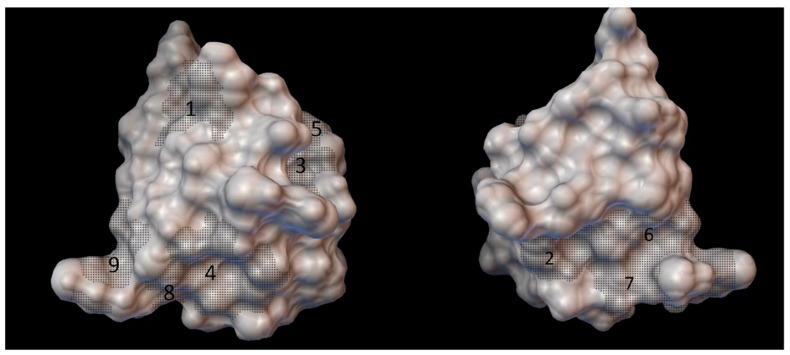
The molecular surface of the insulin molecule from the front and back, marked with small black dots are the 9 binding sites obtained from the CB-Dock, numbered in order of fervency of the parameters which appeared in the results (Marked with 1 the most fervently appeared binding site, and with 9 the most rarely appeared binding site).

**Figure 4 molecules-27-00465-f004:**
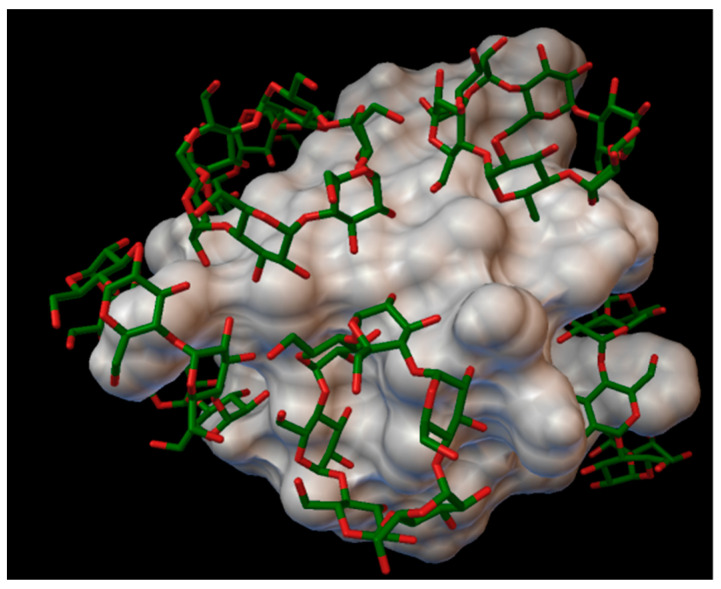
Insulin-α-cyclodextrin complex. The insulin molecular surface is marked in white and the cyclodextrin molecules are marked with with green and red licorice.

**Figure 5 molecules-27-00465-f005:**
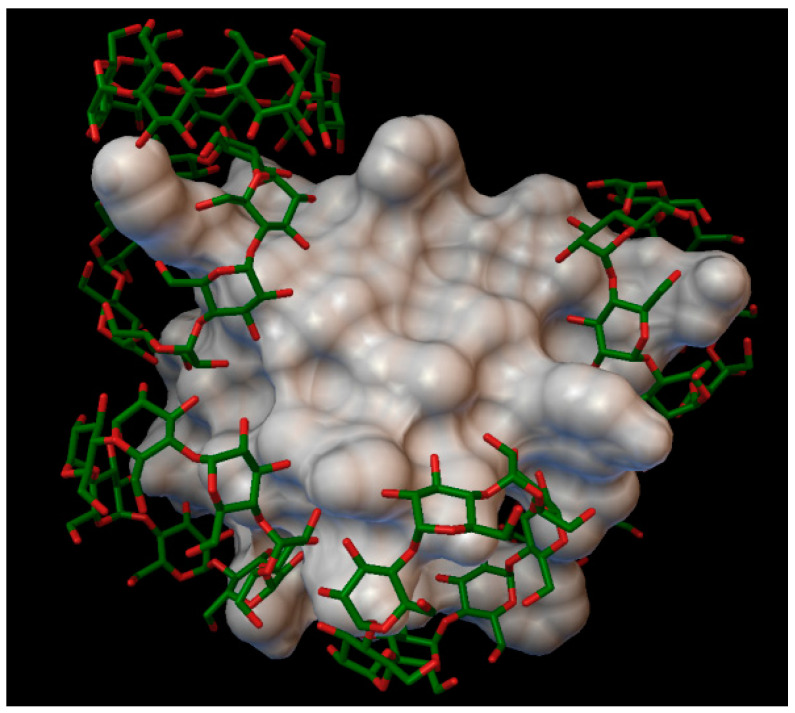
Insulin-β-cyclodextrin complex. The insulin molecular surface is marked with white and the cyclodextrin molecules are marked with green and red licorice.

**Figure 6 molecules-27-00465-f006:**
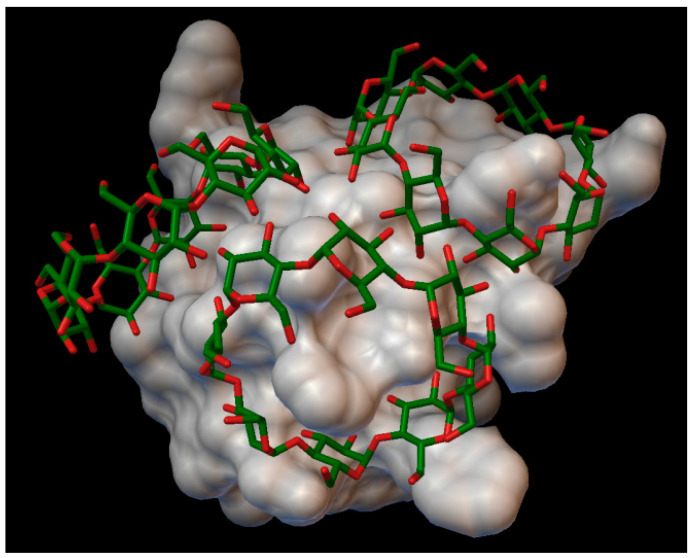
Insulin-γ-cyclodextrin complex. The insulin molecular surface is marked with white, and the cyclodextrin molecules are marked with green and red licorice.

**Figure 7 molecules-27-00465-f007:**
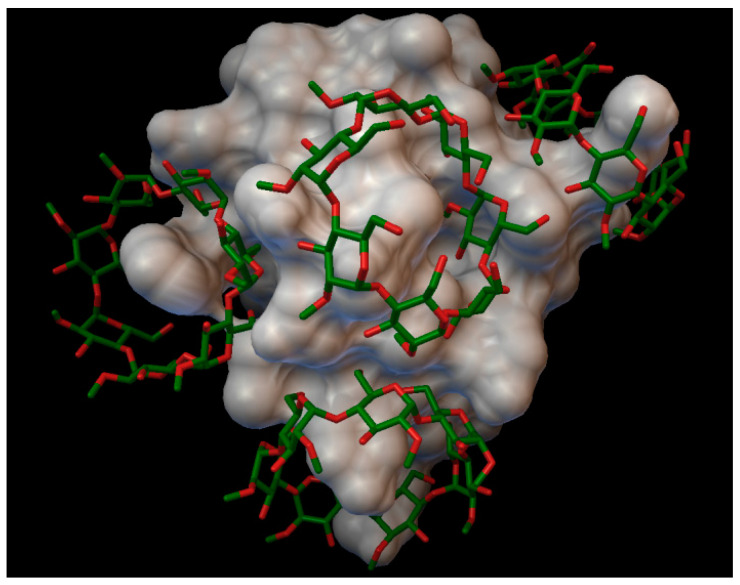
Insulin-methyl-β-cyclodextrin (DS 6) complex. The insulin molecular surface is marked with white and the cyclodextrin molecules are marked with green and red licorice.

**Figure 8 molecules-27-00465-f008:**
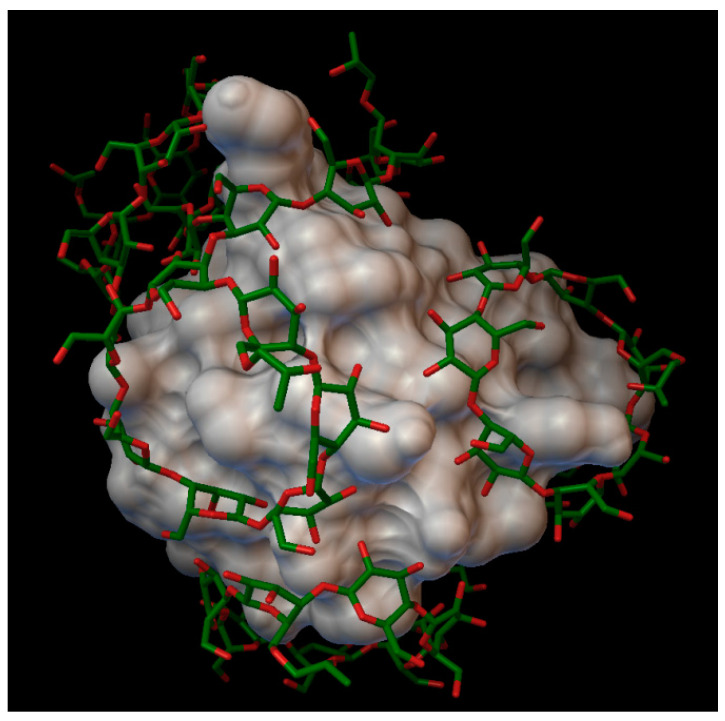
Insulin-hydroxypropyl-β-cyclodextrin (DS 1) complex. The insulin molecular surface is marked in white and the cyclodextrin molecules are marked with green and red licorice.

**Figure 9 molecules-27-00465-f009:**
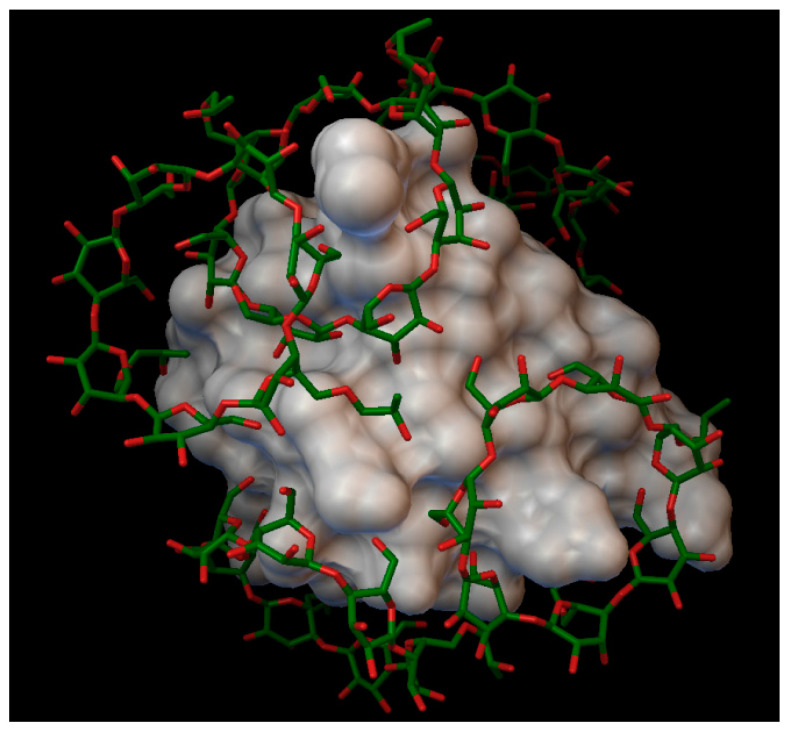
Insulin-hydroxypropyl-β-cyclodextrin (DS 2) complex. The insulin molecular surface is marked with white and the cyclodextrin molecules are marked with green and red licorice.

**Figure 10 molecules-27-00465-f010:**
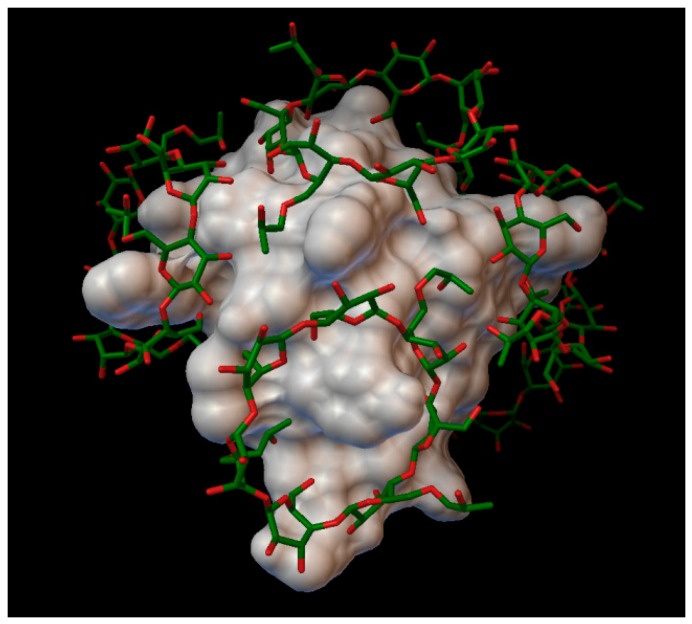
Insulin-hydroxypropyl-β-cyclodextrin (DS 3) complex. The insulin molecular surface is marked with white and the cyclodextrin molecules are marked with with green and red licorice.

**Figure 11 molecules-27-00465-f011:**
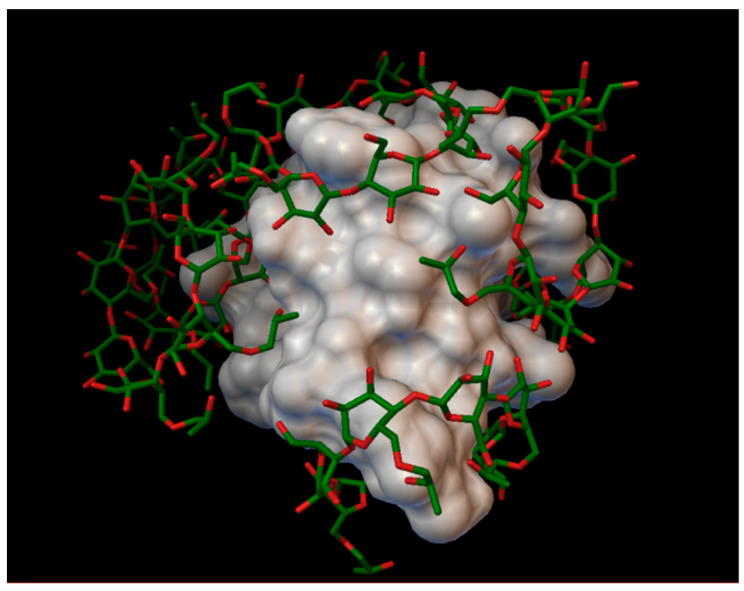
Insulin-hydroxypropyl-β-cyclodextrin (DS 4) complex. The insulin molecular surface is marked with white and the cyclodextrin molecules are marked with with green and red licorice.

**Figure 12 molecules-27-00465-f012:**
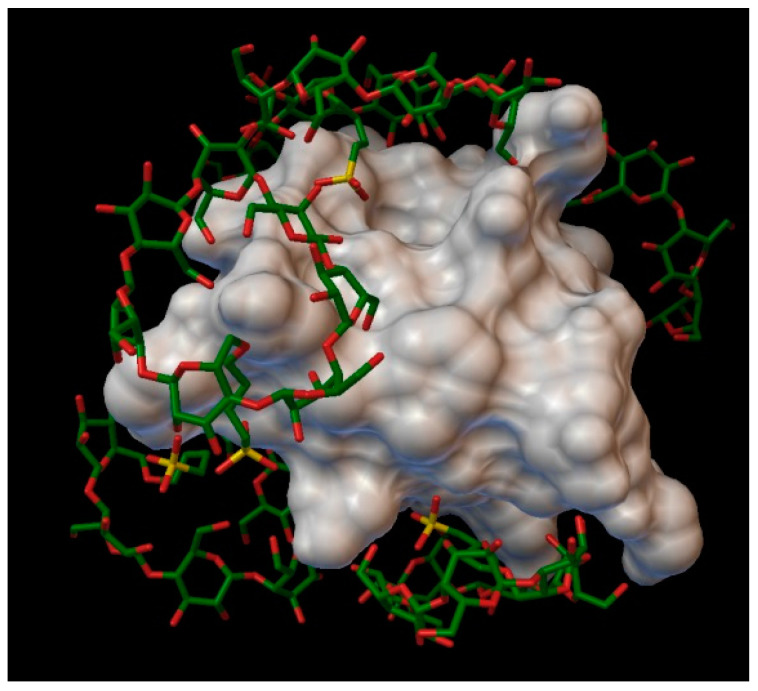
Insulin-sulfobutylether-β-cyclodextrin (DS 1) complex. The insulin molecular surface is marked with white and the cyclodextrin molecules are marked with green and red licorice.

**Figure 13 molecules-27-00465-f013:**
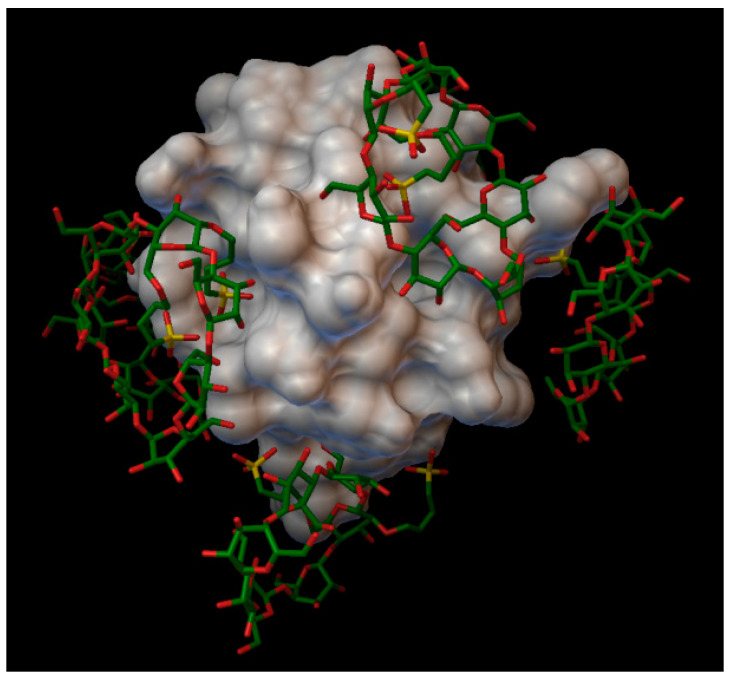
Insulin-sulfobutylether-β-cyclodextrin (DS 2) complex. The insulin molecular surface is marked with white and the cyclodextrin molecules are marked with green and red licorice.

**Figure 14 molecules-27-00465-f014:**
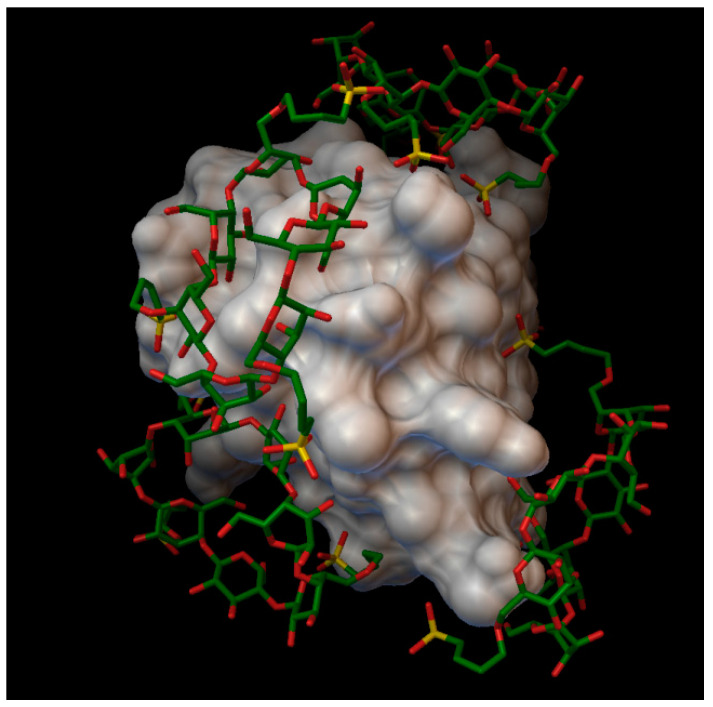
Insulin-sulfobutylether-β-cyclodextrin (DS 3) complex. The insulin molecular surface is marked with white and the cyclodextrin molecules are marked with green and red licorice.

**Figure 15 molecules-27-00465-f015:**
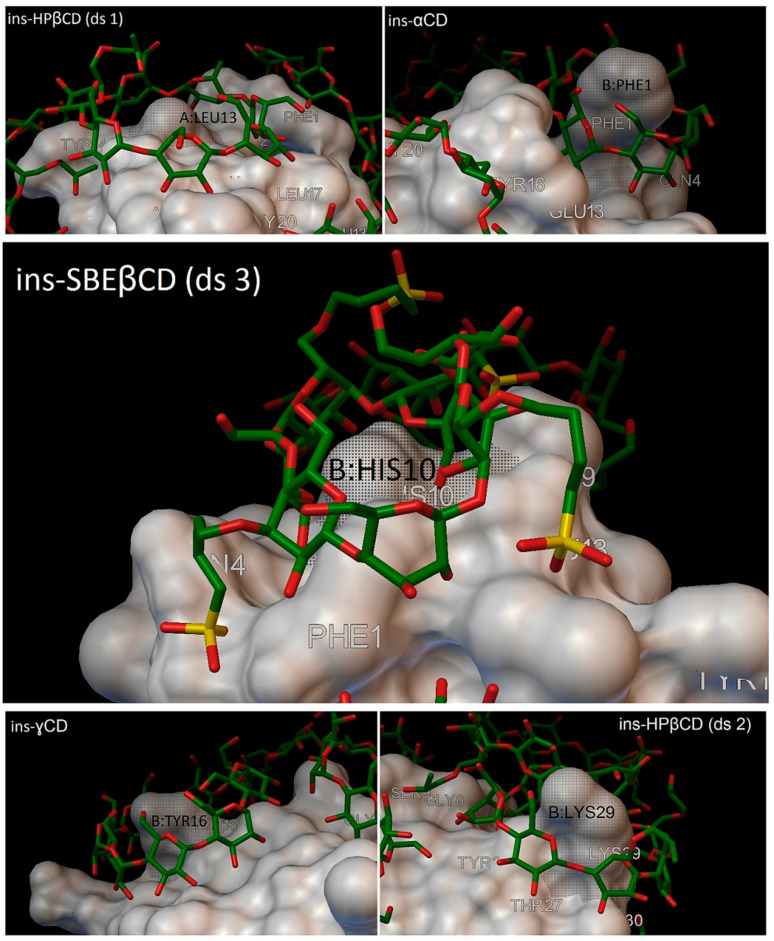
Up, left: residue A:LEU13 entering in the cavity of the hydroxypropyl-β-cyclodextrin (DS 1). Up, right: residue B:PHE1 entering in the cavity of the α-cyclodextrin. Middle: residue B:HIS10 entering in the cavity of the sulfobutylether-β-cyclodextrin (DS 3). Down, left: residue B:TYR16 entering in the cavity of the γ-cyclodextrin. Down, right: residue B:LYS29 entering in the cavity of the hydroxypropyl-β-cyclodextrin (DS 2).

**Figure 16 molecules-27-00465-f016:**
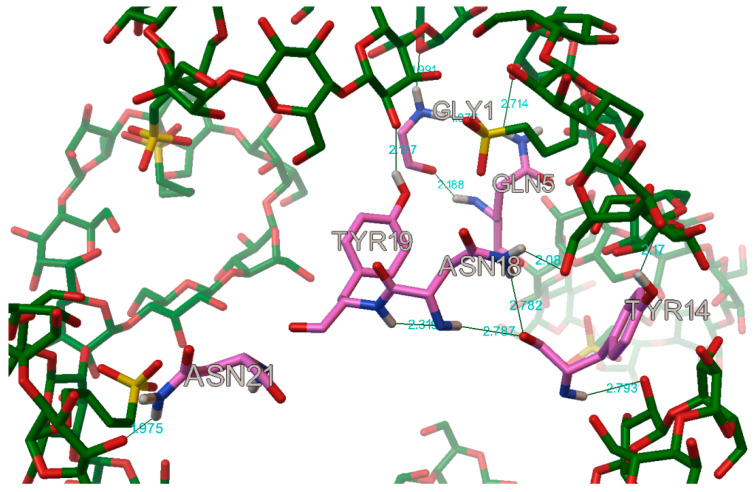
The most commonly occurring hydrogen bond on the A chain in the insulin molecule (in this case with the sulfobutylether-β-cyclodextrin, DS 2).

**Figure 17 molecules-27-00465-f017:**
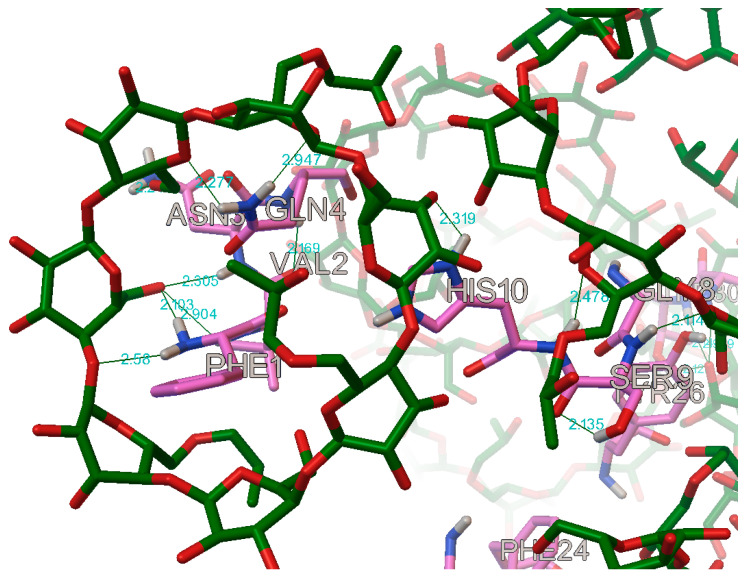
The most commonly occurring hydrogen bonds on the first part of the B chain in the insulin molecule (in this case with the hydroxypropyl-β-cyclodextrin, DS 2).

**Figure 18 molecules-27-00465-f018:**
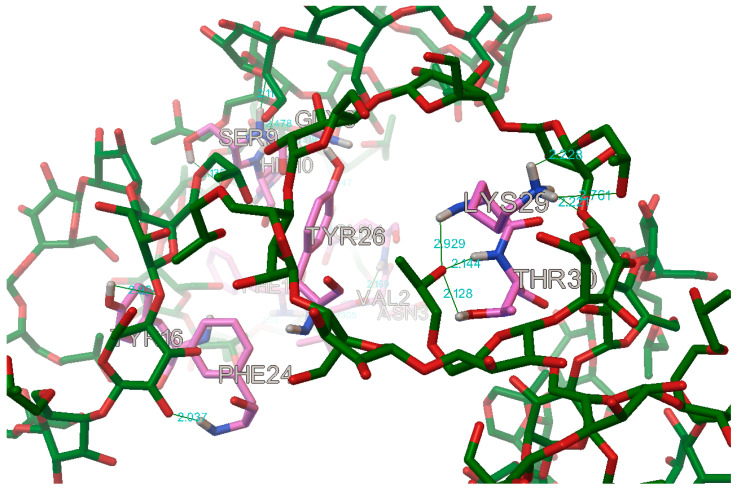
The most commonly occurring hydrogen bonds on the second part of the B chain in the insulin molecule (in this case with the hydroxypropyl-β-cyclodextrin, DS 2).

**Table 1 molecules-27-00465-t001:** The binding energy in the insulin-CD complex.

Binding Energy in the Insulin-CD Complex (kcal/mol)											
CD	1 pos.	2 pos.	3 pos.	4 pos.	5 pos.	6 pos.	7 pos.	8 pos.	9 pos.	Average Binding Energy	Nr of CD in the Complex
α-CD	−9.14	−9.28	−5.51	−7.28	−9.65	−8.23	−10.11	−9.20	−1.74	−8.55	8
β-CD	−9.89	−9.77	−7.59	−9.30	−9.90	−10.30	−9.28	78.93		−9.43	7
γ-CD	−8.18	−9.73	−9.24	−7.64	−4.64					−7.89	5
M-α-CD (DS 12)	−2.19	−0.47	−2.38	0.25	−2.24	−1.13	1.36			−0.97	0
M-β-CD (DS 12)	0.25	0.05	0.14	1.31	1.42	3.58				1.13	0
M-α-CD (DS 18)	−1.55	−1.11	−3.06	1.33	−0.59	−1.19	−0.05			−0.89	0
M-β-CD (DS 6)	−6.56	−7.63	−3.81	−8.59	−6.26	−3.48				−6.06	5
HP-β-CD (DS 1)	−10.81	−9.49	−11.10	−12.83	−11.40	−9.54	−11.07			−10.89	7
HP-β-CD (DS 2)	−4.50	−4.30	−3.91	−6.00	−5.72	−8.49	−5.31			−5.46	7
HP-β-CD (DS 3)	−6.30	−7.47	−6.34	−9.06	−7.90	−4.91				−7.00	6
HP-β-CD (DS 4)	−6.82	−6.23	−5.84	−5.83	−3.98	−1.38	−5.58			−5.09	6
SBE-β-CD (DS 1)	−5.97	−7.09	−9.99	−9.85	−7.94	−12.15	−6.41			−8.49	7
SBE-β-CD (DS 2)	−12.57	−12.44	−11.87	−9.84	−12.33	−18.70				−12.96	6
SBE-β-CD (DS 3)	−7.79	−12.66	−7.99	0.00	−8.66	−16.29				−10.68	5
HE-β-CD (DS 1)	−2.39	−3.05	−1.39	2.68	−0.10	1.11	−1.88			−0.72	0
HE-β-CD (DS 2)	0.12	−0.49	2.39	0.65	0.00	0.98				0.61	0
HE-β-CD (DS 3)	0.60	3.41	−2.12	−0.96	0.64	−3.25				−0.28	0

In this table is displayed the bonding energy of the cyclodextrin in every binding site (position). In the penultimate column is noted the average of the binding energy of the complexes (only the energies under −3.5 kcal/mol were taken in consideration, and in the case of cyclodextrins, where all the energies were over −3.5 kcal/mol, there were calculated the average energy in all the computed positions). In the last column, the number of cyclodextrin which participated in the insulin complex, according to the binding energy (smaller than −3.5 kcal/mol), is summarized. Abbreviations: M—methyl, HP—hydroxypropyl, SBE—sulfobutylether, HE—hydroxyethyl, DS—degree of substitution, pos.—position, CD—cyclodextrin.

## Data Availability

The datasets that support the findings of this study are available from the corresponding authors upon reasonable request.

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
