# Peer review of "Insulin Complexation with Cyclodextrins—A Molecular Modeling Approach"

_molecules, 2022, doi:10.3390/molecules27020465_

Round 1

Reviewer 1 Report

The submitted manuscript entitled "Insulin complexation with cyclodextrins - a molecular modeling approach" is well written, the results are well described and the conclusions are supported by the results. This work can be interesting for those researchers who are working in the field of diabetes mellitus. This reviewer would like the authors to address some minor points before my recommendation for publication in Molecules:

1. The expressions "in silico" and "in vitro" should be in italic format

2. Define the abbreviation WHO

3. Line 33: replace "witch" by "with"

4. Line 66: replace "storing stage" by "storage stage"

5. Line 77: replace "from" by "form"

6. Line 137: replace "bond" by "bind"

Author Response

Point 1: The expressions "in silico" and "in vitro" should be in italic format

Response 1: Thank you, we changed the format.

Point 2: Define the abbreviation WHO

Response 2: The WHO is the abbreviation of the World Health Organization. In the article we changed to the whole name: World Health Organization.

Point 3: Line 33: replace "witch" by "with"

Response 3: Replaced.

Point 4: Line 66: replace "storing stage" by "storage stage"

Response 4: Replaced.

Point 5: Line 77: replace "from" by "form"

Response 5: Replaced.

Point 6: Line 137: replace "bond" by "bind"

Response 6: Replaced.

Reviewer 2 Report

Dear Authors

The paper described in silico investigation of complexation between insulin and cyclodextrin. The data presented was only binding energy obtained from blind docking simulation with AutoDock. There are some suggestions before the paper accepted:

  1. The method presented is relatively simple in term only using docking method. Is this procedure new and no one reported the same method?
  2. How to know that the binding energy obtained is true? Is there any validation for the docking protocol?
  3.  Do the study determine the stablity of the complex? With the influencing factors such as storage time and temperature?

Author Response

Point 1: The method presented is relatively simple in term only using docking method. Is this procedure new and no one reported the same method?

 Response 1: This method is partially new. The Cavity-detection guided Blind Docking, the short docking and the flexible docking are common docking methods. The steps described in the Method, Flexible docking section is our invention. We find an accurate, cost-efficient procedure to dock more than one ligand (theoretically between 5-10) with a peptide molecule, for predicting the molar ratio and the conformation of the complex. This method is detailed in the Method section.

Row 126: In addition, to clarify the Reviewer’s comment, a paragraph was introduced: “For predicting the molar ratio and the conformation of the complex, we designed an ac-curate, cost efficient procedure to dock the insulin with more than one ligand (theoretically between 5-10, according to the literature [11]), but we also kept the computational de-mands at a reasonable level. This method is detailed in the Method section.”

Point 2: How to know that the binding energy obtained is true? Is there any validation for the docking protocol?

 Response 2: The AutoDock program master equation is a well-defined, precise equation, which is based on many computational research developed by many scientists between 1950-1990 (in our research, in the introduction part we mention the components of the function, with bibliography). The AutoDock program at the first time were tested, and calibrated in 1998, by using 30 structurally known complexes with experimentally determined binding constants. (Morris, G. M.; Goodsell, D. S.; Halliday, R. S.; Huey, R.; Hart, W. E.; Belew, R. K.; Olson, A. J. Automated Docking Using a Lamarckian Genetic Algorithm and an Empirical Binding Free Energy Function, J. Comput. Chem. 1998, 19 (14), 1639-1662.) In our bibliography between 19-24 are placed the reference for the docking protocol. Since the appearance of the AutoDock program yearly more and more molecular modeling article appear using the AutoDock software for docking the protein complexes. More than 7000 publication is registered on the PubMedCentral, using the AutoDock for docking method and over 30,000 citations are reported by Google Scholar (Goodsell DS, Sanner MF, Olson AJ, Forli S. The AutoDock suite at 30. Protein Sci. 2021 Jan;30(1):31-43. doi: 10.1002/pro.3934. Epub 2020 Sep 12. PMID: 32808340; PMCID: PMC7737764.)

Row 99: To explain in the manuscript the topic presented at this point, a sentence was introduced: “More than 7000 publication is registered on the PubMed Central, using the AutoDock for docking method and over 30,000 citations are reported by Google Scholar [46].”

Point 3: Do the study determine the stability of the complex? With the influencing factors such as storage time and temperature?

 Response 3: The AutoDock program is used to determine the binding energies of the complexes. The binding energy represents the stability of the complexes, which is inversely proportionate with the stability of the complex: smaller binding energy representing a greater stability. Yes, the study determines the stability of the complex. This study is the first step of a bigger research, where we will prepare the complexes using different methods and will determinate the physico-chemical properties of the prepared complexes (stability constant, the influence of the storage time and temperature on the complex stability, etc.). This step is only used to determine the best cyclodextrin to use in the next step, is a cost-efficient way, so we don`t need to prepare and test 18 different complexes to find the best, instead we determine the stability by docking, and by comparing the result, we can choose 1 or 2 complexes with the best results to test out.